# *vwa1* Knockout in Zebrafish Causes Abnormal Craniofacial Chondrogenesis by Regulating FGF Pathway

**DOI:** 10.3390/genes14040838

**Published:** 2023-03-30

**Authors:** Xiaomin Niu, Fuyu Zhang, Lu Ping, Yibei Wang, Bo Zhang, Jian Wang, Xiaowei Chen

**Affiliations:** 1Department of Otolaryngology, Peking Union Medical College Hospital, Chinese Academy of Medical Sciences and Peking Union Medical College, Beijing 100730, China; 28-Year MD Program, Chinese Academy of Medical Sciences and Peking Union Medical College, Beijing 100730, China; 3Department of Otolaryngology-Head & Neck Surgery, China-Japan Friendship Hospital, Beijing 100730, China; 4Key Laboratory of Cell Proliferation and Differentiation of the Ministry of Education, College of Life Sciences, Peking University, Beijing 100730, China

**Keywords:** hemifacial microsomia, *VWA1*, FGF pathway, cranial neural crest cells

## Abstract

Hemifacial microsomia (HFM), a rare disorder of first- and second-pharyngeal arch development, has been linked to a point mutation in *VWA1* (von Willebrand factor A domain containing 1), encoding the protein WARP in a five-generation pedigree. However, how the *VWA1* mutation relates to the pathogenesis of HFM is largely unknown. Here, we sought to elucidate the effects of the *VWA1* mutation at the molecular level by generating a *vwa1*-knockout zebrafish line using CRISPR/Cas9. Mutants and crispants showed cartilage dysmorphologies, including hypoplastic Meckel’s cartilage and palatoquadrate cartilage, malformed ceratohyal with widened angle, and deformed or absent ceratobranchial cartilages. Chondrocytes exhibited a smaller size and aspect ratio and were aligned irregularly. In situ hybridization and RT-qPCR showed a decrease in *barx1* and *col2a1a* expression, indicating abnormal cranial neural crest cell (CNCC) condensation and differentiation. CNCC proliferation and survival were also impaired in the mutants. Expression of FGF pathway components, including *fgf8a*, *fgfr1*, *fgfr2*, *fgfr3*, *fgfr4*, and *runx2a*, was decreased, implying a role for VWA1 in regulating FGF signaling. Our results demonstrate that VWA1 is essential for zebrafish chondrogenesis through effects on condensation, differentiation, proliferation, and apoptosis of CNCCs, and likely impacts chondrogenesis through regulation of the FGF pathway.

## 1. Introduction

Hemifacial microsomia (HFM (MIM:164210)), also termed craniofacial microsomia or oculo-auricular–vertebral spectrum (OAVS), is a rare congenital developmental disorder affecting derivatives of the first and second branchial arches with an estimate incidence of 1 in 5600 live births [1]. The heterogeneous etiologies of the disease include both environmental and genetic factors [2]. Although genes, including *OTX2*, *ITGB4*, *PDE4DIP*, and *FRK,* have been linked with the disease, the majority of HFM cases still lack an identifiable underlying genetic cause [3,4,5].

During embryogenesis of the outer ear, cranial neural crest cells (CNCCs) migrate to the mesenchyme of the first and second pharyngeal arches, forming the connective tissue, pericytes, and smooth muscle cells of blood vessels [6]. A series of studies have identified dysregulated CNCC and cartilage development as the potential causes of HFM and related syndromes [7,8,9]. The fibroblast growth factor (FGF) signaling pathway has been demonstrated to play vital roles in chondrogenesis during the early stages of development, exhibiting intimate crosstalk with tumor growth factor-β (TGFβ) and Wnt (wingless-related integration site) signaling pathways [10]. Previous murine models have demonstrated that elements of the FGF pathway are required for craniofacial development, including fibroblast growth factor receptor 1 (FGFR1), fibroblast growth factor 8 (FGF8), and FGF4 [11].

In our previous work [12], we identified a point mutant of VWA1 (c.905G>A) as a candidate pathogenic HFM gene by whole exome sequencing (WES) in a five-generation pedigree. Additionally, we used the zebrafish model for further investigation of the function of VWA1 on craniofacial development. The signaling patterns and anatomical features of zebrafish’s first pharyngeal arch development were evolutionarily homologous with mammals, and gene editing can be easily done in zebrafish with their craniofacial cartilage morphology to be clearly visualized [13]. Therefore, zebrafish is regarded as a promising model for understanding craniofacial anomalies and has been used in several studies to elucidate the role of some pathogenic genes for microtia-related syndromes [14,15,16]. Using morpholino to knock down the homologous gene, vwa1, in zebrafish, we demonstrated that the gene product, von Willebrand factor A-domain related protein (WARP), contributes to the development of craniofacial cartilage by supporting both the proliferation of CNCCs and the organization of pharyngeal chondrocytes. WARP is predominantly expressed in the extracellular matrix (ECM) of muscle, peripheral nerves, and chondrocytes [17], where its expression overlaps with that of collagen VI [18]. Functionally, WARP appears to stabilize the ECM structure by linking collagen VI to perlecan, a basement membrane proteoglycan [19] and thus plays a critical role in chondrogenic structure and function [12]. 

However, little is known about the molecular mechanisms linking *VWA1* to HFM, and our previous attempt to address this issue using morpholinos [12] to knock down the orthologous gene, *vwa1*, in zebrafish was inevitably limited by off-target effects, short maintenance time, and non-specific activation of the P53 pathway. An alternative to the morpholino approach is CRISPR/Cas9-based gene editing, which offers the advantages of precise targeting, low off-target effects, low cytotoxicity, and the ability to generate stably inherited mutant fish lines. Therefore, to further study the function of vwa1 related to craniofacial cartilage development and malformation, in the current study, we generated *vwa1*-knockout zebrafish using the CRISPR/Cas9 technique. We demonstrated that *vwa1* mutants showed craniofacial cartilage deformities with hypoplastic Meckel’s cartilage; palatoquadrate, ceratohyal, and ceratobranchial cartilages; and disordered arrangement of chondrocytes at 4 days post fertilization (dpf). We further found that vwa1 regulates the proliferation and differentiation of CNCCs, as well as the arrangement of chondrocytes in the zebrafish mandible. In addition, we showed that the downregulation of components of the FGF pathway was involved in the maldevelopment of craniofacial cartilages.

## 2. Materials and Methods

### 2.1. Zebrafish and Embryos

Zebrafish (*Danio rerio*) used in this study were Tuebingen and the transgenic line, *Tg(sox10: EGFP)*. All zebrafish embryos were raised under standard temperature (28.5 °C) and light (14 h/10 h light/dark cycle) conditions and were staged based on hours post fertilization (hpf) [13]. Where necessary, 1-phenyl-2-thiourea (0.002%; Sigma, Beijing, China, P7629) was added to the embryo media to prevent pigment development.

### 2.2. Zebrafish vwa1 Mutant Generation

For *vwa1* knockout in zebrafish using CRISPR/Cas9, four CRISPR target sites were designed with the online software CHOPCHOP, and primers containing a T7 promoter sequence were synthesized. Guide RNA (gRNA) templates were synthesized by polymerase chain reaction (PCR) using T7-vwa1-sfd and tracr rev as primers and the pMD19-gata5 plasmid as the template. The PCR-amplified plasmid region (WT sequence: 5′-GTT TTA GAG CTA GAA ATA GCA AGT TAA AAT AAG GCT AGT CCG TTA TCA ACT TGA AAA AGT GGC ACC GAG TCG GTG CT-3′) was sequenced using the primers RV-M (forward) and M13-17 (reverse). The PCR product was purified and transcribed in vitro to gRNAs using the T7 RiboMAX Express Large Scale RNA Production System (Promega, Beijing, China, P1320). Cas9/gRNAs complexes were formed by incubating gRNAs with 600 ng/μL Cas9 (Novoprotein, Beijing, China, E365) at 37 °C for 15 min and then microinjected into the animal pole of the embryos at the one-cell stage. The efficacy of gRNAs was verified by PCR and Sanger sequencing of crude genomic DNA extracts from the embryos. F0 somatic mosaic embryos (crispants) in the same batch were raised to adulthood and screened for the presence of germline-transmitted *vwa1* indels. The target sequences and primer sequences used to amplify the target region are shown in Appendix A.

### 2.3. Reverse-Transcription Quantitative Polymerase Chain Reaction (RT-qPCR)

The expression of *vwa1*, *fgfr1a*, *fgfr2*, *fgfr3*, *fgfr4*, *fgf8a*, *fgf8b*, and *runx2a* in collected embryos (n = 30) was determined by RT-qPCR using the primers shown in Appendix A. Total RNA was extracted with TRIzol (Invitrogen, Beijing, China) and purified to an A_260_/A_280_ ratio of 1.8–2.0, after which 1 μg of each sample was reverse transcribed using All-In-One 5X RT MasterMix (Abm, Beijing, China, G592). qPCR was then performed using the UltraSYBR Mixture (CWBIO, CW0957H), and expression levels of target genes were quantified using the 2^−ΔΔCT^ method and normalized to β-actin expression levels. Expression of mRNA for *vwa1*-mutants is presented as fold-change relative to that of wild-type (WT) controls.

### 2.4. Whole-Mount In Situ Hybridization (WISH)

The PCR products of *crestin*, *dlx2a, tbx1*, *barx1*, *nkx2.3*, *fgf8a*, *fgfr2, fgfr3, sox9a*, and *col2a1a* were cloned into the *pEASY*-T3 Cloning Vector (TransGen, Beijing, China, CT301-01) using the primers shown in Appendix A. after which the plasmids were sequenced and linearized. RNA probes for in situ hybridization were synthesized using digoxigenin (DIG) RNA labeling mix (Roche, Beijing, China, 11277073910) and T7 polymerase (Promega, p4074) and dissolved in RNase-free water. Whole-mount in situ hybridization (WISH) was performed as previously described [14]. Briefly, embryos were collected and fixed in 4% paraformaldehyde (PFA) overnight at 4 °C. After dehydration with an ethanol series, embryos were stored at −20 °C. Embryos were rehydrated in a series of ethanol/PBST solution (PBS + 0.1% Tween-20), permeabilized with protease K (TransGen, GE201-01), then fixed with 4% PFA at room temperature for 30 min and rinsed again with PBST solution. After pre-hybridization for 3 h in hybridization solution containing 50% formamide (Sigma, F9037) at 65 °C, hybridization was performed overnight at 65 °C using antisense probes diluted to 2–3 ng/μL in hybridization solution. Embryos were sequentially rinsed at 65 °C with a series of mixtures of hybridization solution/2× SSCT buffer at different ratios, ending with 2× SSCT. Embryos were then washed with 0.2× SSCT solution at room temperature, followed by a wash with PBST solution. After blocking in blocking solution, consisting of 0.2% bovine serum albumen (BSA) and 0.1% Tween-20 in PBS, for 3 h at room temperature, embryos were incubated with anti-DIG-AP-antibody (1:4000; Roche, 11093274910) at 4 °C overnight. Finally, embryos were rinsed six times with PBST solution for more than 15 min each and then stained with NBT/BCIP solution (Coolaber, Beijing, China, SK2030-125 mL). WISH images were captured using a stereomicroscope (Stemi 305; Zeiss, Jena, Germany) equipped with AxioCam 208 color (Zeiss) and processed using ZEN 3.1 software (blue edition).

### 2.5. Cartilage Staining

Embryos were collected at 4 dpf and fixed by incubating with 4% PFA at 4 °C overnight. Cartilage in embryos was visualized by Alcian Blue staining of cartilage and wheat germ agglutin (WGA) staining of chondrocyte membranes. Alcian Blue cartilage staining was performed according to a previously reported standard protocol [15]. For WGA chondrocyte membrane staining, embryos were fixed in 4% PFA overnight and then extensively washed in PBST. Embryos were then stained with Alexa Fluor 594-conjugated WGA (ThermoFisher, Beijing, China, W11262), diluted 1:200 by incubating overnight at 4 °C, and again washed with PBST [16]. Embryos were mounted in 1.5% low-melting glue, and the arrangement of pharyngeal chondrocytes was observed by confocal microscopy.

### 2.6. Terminal Deoxynucleotidyl Transferase dUTP Nick End Labeling (TUNEL) 

TUNEL assays were performed using the In Situ Cell Death Detection Kit, TMR Red (Roche, 12156792910). WT and *vwa1* mutant embryos were collected, fixed with 4% PFA at 4 °C overnight, washed with PBST, and permeabilized by treating with proteinase K. Permeabilized embryos were fixed in 4% PFA for 30 min at room temperature, then washed three times with PBST and incubated overnight at 4 °C with TUNEL labeling mixture. After washing with PBST, embryos were mounted in 1.5% low-melting glue and observed under a confocal microscope.

### 2.7. Whole-Mount Immunofluorescence Staining

Whole-mount immunofluorescence staining was performed as previously reported [17,18]. Briefly, WT and *vwa1* mutant embryos were collected and fixed by incubating overnight with 4% PFA at 4 °C. Samples were rinsed three times each in PBST and double-distilled water and then permeabilized by incubating with acetone for 10 min at −20 °C. After again rinsing in double-distilled water and PBST, embryos were blocked by incubating at room temperature for 3 h in a blocking solution consisting of 10% goat serum, 10% BSA (Sigma, B2064), 0.5% Triton X-100, and 1% DMSO (dimethyl sulfoxide) in PBS. Next, embryos were incubated with primary anti-p-histone H3 (C-2) mouse polyclonal antibody (Santa Cruz Biotechnology, sc-374669), diluted 1:200 in blocking solution overnight at 4 °C with gentle shaking. The primary antibody was then removed, and embryos were washed four more times with blocking solution for at least 1 h each. Finally, embryos were incubated overnight at 4 °C in blocking solution containing Alexa Fluor 594-conjugated goat–anti-mouse IgG (H+L) cross-adsorbed secondary antibody (1:400; Invitrogen, A11005). DAPI (4′,6-diamidino-2-phenylindole; Beyotime, C1002), diluted 1:1000, was used to visualize nuclei. After washing, embryos were fixed in 1.5% low-melting glue, and immunostained images were observed and recorded at the same settings using Nikon A1 confocal microscopy and Zeiss Axio Scan.

### 2.8. Statistical Analyses

All experiments were performed in triplicate, and statistical significance was assessed by unpaired *t*-tests using GraphPad Prism 6.0 (GraphPad Software, San Diego, CA, USA), with *p*-values < 0.05 considered statistically significant. Statistical analyses of images were performed using ImageJ v1.8.0. All phenotypic observational experiments showed congruent results, affecting >70% of embryos observed, and all selected images are representative.

## 3. Results

### 3.1. Generation of a vwa1-Knockout Zebrafish Line Using CRISPR/Cas9

The evolutionarily conserved human *VWA1* and zebrafish *vwa1* genes encode a protein containing a von Willebrand factor A-domain and two fibronectin type III repeats. A multiple sequence alignment demonstrated an overall amino acid identity between human and zebrafish VWA1 of 35.5% (Figure 1A), with domain-specific identity varying from 27.8% and 36.5% for the two fibronectin type III repeat domains to 46.5% for the von Willebrand factor A-domain.

To verify the role of VWA1 in mandibular development, we knocked out *vwa1* in zebrafish embryos using CRISPR/Cas9 and studied the effects of both somatic mosaic and germinal VWA1 loss of function in zebrafish. Four gRNAs were designed to target protospacer adjacent motif (PAM) sites in exons 2, 3, and 4 (Figure 1B). After in vitro transcription of gRNA, the Cas9/gRNA complex was injected into one-cell–stage zebrafish embryos (Figure 1C). Sequencing of *vwa1*-knockout (KO) zebrafish embryos guided by the four gRNAs revealed the presence of multiple peaks downstream of the PAM site, indicating various indels, in three gRNAs, whereas sequencing of embryos generated by the second gRNA targeting exon 2 failed because the target was located near the intron (Figure 1D). Sequential validation of gRNAs revealed that targeting exons 3 and 4 was the most efficient. Therefore, F0 offspring (crispants) were raised by injecting gRNA against exons 3 and 4, respectively. To examine germline transmission, we outcrossed adult F0 zebrafish with WT zebrafish (Figure 1C). F1 fish carrying the mutation were raised to adulthood for subsequent matings to obtain F2, and their fins were clipped for sequencing. F1 mutant adults were crossed to produce F2 homozygous (vwa1^−/−)^ embryos. Unfortunately, in the F2 offsprings of embryos targeted at exon 3, phenotypic inconsistency was observed constantly in embryos carrying no *vwa1* mutation, which we suspected was a result of background mutation. Therefore, the gRNA targeting exon 4 was utilized for our experiments, and the offspring were validated by sequencing (Figure 1E,F).

### 3.2. vwa1 Deletion in Zebrafish Causes Variable Craniofacial Defects

Compared with WT controls, CRISPR/cas9-microinjected *vwa1* crispants and homozygous embryos showed multiple morphological changes, including mandibular dysplasia, pericardial edema, and delayed or absent swim bladder development, that progressively worsened from 3 to 5 dpf (Figure 2A). In addition, based on our observation, *vwa1*^−/−^ embryos were of variable sizes. Some were smaller in size from 30 hpf to 5 dpf compared with WT, but there was no statistical size difference between the two groups in general (Appendix A). To better elucidate the abnormalities caused by *vwa1* knockout, we selected WT and mutant fish with similar sizes for the subsequent experiments.

Alcian Blue staining of embryos at 5 dpf revealed that both crispants and germinal mutants displayed severe craniofacial cartilage defects. Individual cartilage elements exhibited hypoplasia, including hypoplastic Meckel’s cartilage and palatoquadrate cartilage, malformed ceratohyal with widened angle, and deformed or even absent ceratobranchial cartilages (Figure 2B). We validated the knockout by performing quantitative RT-PCR (RT-qPCR) and found that the concentration of *vwa1* mRNA was significantly decreased in both crispants and mutant embryos with a mandibular malformation (Figure 2C), likely owing to nonsense-mediated mRNA decay [20,21]. Together, these results suggest that the deletion of *vwa1* interferes with the process of mandible development in zebrafish.

### 3.3. vwa1 Mutation Leads to Disarrangement of Chondrocytes

Because VWA1 is mainly located in the extracellular matrix of chondrocytes, we stained chondrocyte membranes with wheat germ lectin (WGA) to observe the effects of *vwa1* loss on the arrangement of embryonic mandibular chondrocytes. WT *Tg(sox10: EGFP)* embryos exhibited thin, elongated chondrocytes that assembled closely in a “stacked-coin” manner (Figure 3A,B). In contrast, chondrocytes in *vwa1*^−/−^ embryos were smaller with a smaller aspect ratio and were aligned irregularly (Figure 3C,D). These results suggest that hypoplasia of the mandible observed after the knockout of *vwa1* is likely attributable to the abnormal alignment of chondrocytes.

### 3.4. Chondrogenesis Is Impaired in Association with Abnormal CNCC Condensation and Differentiation in vwa1^−/−^ Zebrafish

To elucidate the mechanism through which VWA1 loss of function causes pharyngeal cartilage defects, we assessed the expression of several marker genes by whole-mount in situ hybridization (WISH). During the development of zebrafish pharyngeal cartilage, the induction of CNCCs could be marked by *crestin* at 24 hpf. CNCCs then migrate to the pharyngeal arches, where they subsequently (30 hpf) differentiate into pharyngeal arch primordia and eventually become mature. When the CNCCs migrate into the primordia, they are marked by *dlx2a* [22]. Furthermore, *tbx1* marks the mesoderm core and endoderm of the primordia that are not derived from CNCCs [23]. *nkx2.3*, on the other hand, marks the endoderm of the developing pharyngeal arches [24]. CNCCs in the primordia subsequently aggregate and differentiate, which are respectively marked by the expression of *barx1* at 48 hpf and *sox9a* at 60 hpf. Differentiated chondrocytes, identified by their expression of *col2a1a*, are organized into chondrocyte stacks from 72 hpf to 84 hpf to form pharyngeal cartilage, including the mandible.

WISH revealed that the expression of *crestin* was similar in WT controls and mutants (Figure 4A), indicating that VWA1 is not necessary for the induction of CNCCs. The expression area of *dlx2a* and *tbx1* in mutants was smaller in absolute terms, but the expression pattern was conserved relative to the smaller body size of these mutants (Figure 4B). The expression of *nkx2.3* was also normal in *vwa1* mutants (Figure 4C). Thus, in *vwa1* mutants, the pharyngeal arch primordia developed properly, and CNCCs migrated normally. However, *barx1* expression in the mutants was significantly reduced, indicating that the functional loss of VWA1 impaired the condensation of CNCCs. Furthermore, the expression of *sox9a* in *vwa1* mutants was similar to that in controls, whereas the expression of *col2a1a* was significantly reduced. Therefore, we speculate that the functional loss of VWA1 contributes to the malformation of pharyngeal cartilage by disrupting the condensation and differentiation of CNCCs.

### 3.5. The Proliferation of CNCCs Required vwa1

In addition to condensation defects in the pharyngeal arches of *vwa1* mutants, the reduced size of palatoquadrate and ceratohyal cartilages also indicates potential disruption of CNCC proliferation and/or survival. To assess CNCC apoptosis during development, we performed TUNEL assays on embryos at 24, 30, and 48 hpf. Confocal imaging revealed a higher rate of apoptosis at 24 hpf in *vwa1* mutants than in WT controls. A certain amount of apoptotic cells, predominantly in the dorsal part of the embryos, was observed in *vwa1* mutants (Figure 5A,B). In contrast, TUNEL assays showed an absence of apoptotic cells in the pharyngeal region of *vwa1* mutants at 30 and 48 hpf (Appendix A). To assess CNCC proliferation at 30 hpf, we also immunostained Tg(sox10: GFP) embryos for phosphorylated histone H3 (PHH3). PHH3 staining revealed a notable reduction in mitotic PHH3-positive CNCCs in *vwa1* mutant embryos compared with control embryos at 30 hpf, indicating inhibition of CNCC proliferation (Figure 5C,D).

### 3.6. vwa1 Regulates FGF Signaling to Modulate the Development of Mandibular Cartilage

We demonstrated that, among the signaling pathways closely related to craniofacial development, FGF signaling might interact with VWA1. At present, 22 FGFs and five FGF receptors (FGFRs) are known in humans. The binding of an FGF ligand to an FGFR induces receptor dimerization, which leads to transphosphorylation and subsequent activation of a number of downstream cascades, including Ras/MAP kinase, phospholipase Cγ, PI3-kinase and STAT [21,22]. The FGF pathway regulates developmental processes such as cell proliferation, migration, and differentiation—all of which are necessary for bone and cartilage formation [23].

To examine whether the inactivation of vwa1 affects FGF signaling, we measured the mRNA levels of FGF signaling pathway components in *vwa1* mutants and WT controls. We found that the expression of *fgf8a* and FGF receptors (*fgfr1*, *fgfr2*, *fgfr3,* and *fgfr4*), as well as the downstream signaling molecule *runx2a,* was decreased in the *vwa1*^−/−^ zebrafish embryos, whereas the expression of *fgf8b* was not significantly different (Figure 6A). WISH staining showed reductions in the expression of *fgf8a* at 48 hpf and *fgfr2* and *fgfr3* at 4 dpf in mutant embryos (Figure 6B). These data indicate that vwa1 regulates FGF signaling and further affects the development of mandibular cartilage.

## 4. Discussion

In our previous work, a point mutation of *VWA1* was predicted as a candidate gene for HFM, which causes craniofacial deformities. To verify the role of *VWA1* in mandible development, we used CRISPR/Cas9 to generate a *vwa1*^−/−^ zebrafish line by targeting a PAM site in exon 4. The resulting mutants exhibited a series of craniofacial cartilage dysmorphologies in association with abnormal chondrocytes. Further investigations of chondrogenesis showed that CNCC condensation, differentiation, apoptosis, and proliferation were impaired in mutants. The expression of *fgf8a*, its receptors, and downstream effectors were also decreased in the mutants, indicating a role for vwa1 in the regulation of the FGF pathway. Thus, *vwa1* was essential for the development of zebrafish craniofacial cartilages by virtue of its effects on the condensation, differentiation, apoptosis, and proliferation of CNCCs and regulation of the FGF pathway.

Zebrafish *vwa1* is homologous to human *VWA1*, with both encoding WARP, containing a von Willebrand factor A-domain and two fibronectin type III domains. In this study, *vwa1* knockout targeting exon 2 showed low efficiency, whereas zebrafish in which exon 3 was targeted simultaneously acquired background mutations. *vwa1*^−/−^ mutants were generated using a gRNA against exon 4, resulting in a frameshift mutation and early termination. Our results demonstrated that knockout of *vwa1* affected both the structure and the expression of the gene, possibly via nonsense-mediated mRNA decay, and thus contributed to the phenotypic changes of the fish.

*vwa1*-KO crispants and germinal mutants shared craniofacial cartilage phenotypes similar to those of *vwa1* mutants generated in previous studies [12,25]. Both mutants generated by CRISPR/Cas9 and those generated using morpholinos showed craniofacial defects in hypoplastic Meckel’s cartilages and palatoquadrate cartilages and a larger angle between ceratohyal cartilages. Furthermore, the chondrocytes of these mutants exhibited an irregular arrangement and were smaller in size and aspect ratio. Taken together, these similar phenotypes reaffirm that vwa1 is required for the development of craniofacial cartilages. We also generated a *vwa1* mutant fish line and used it to exclude potential influences of gRNA injection on cartilage development.

However, compared with the morphants [12], some crispants and germinal mutants developed ceratobranchial cartilages, albeit deformed. Since morpholino injection in zebrafish could activate p53 and induce cell death [26], it may be that more chondrocytes in morphants experience apoptosis than those in mutants generated by CRISPR/Cas9. Therefore, the cartilages of morphants may be smaller in size and weaker in strength, resulting in more severe deformities. In addition to the advantages of low off-target effects and reduced cytotoxicity, CRISPR/Cas9 also excels in specificity [27] and maintenance time, making our results more compelling.

The phenotypes of *vwa1* mutants are likely caused by impaired CNCC condensation, differentiation, proliferation, and apoptosis. CNCCs are multipotent stem cells that originate from the neural plate border [28], and their development can generally be divided into induction, migration, and differentiation [29]. During gastrulation and neurulation, CNCCs between the neural and non-neural ectoderm are induced and then give rise to mesenchymal cell types of facial cartilage and bone [30,31]. CNCC condensation is critical in establishing the size and shape of the cartilage and is an essential prerequisite for chondrogenesis [32]. After condensation, CNCCs differentiate and express *col2a1a* encoding collagen II. As the scaffold for cartilage ECM, collagen II is crucially involved in signal transduction between chondrocytes and the matrix, regulating cartilage homeostasis. Reduced *col2a1a* expression in zebrafish causes craniofacial cartilage dysmorphologies [33], and *col2a1a* mutations can cause skeletal dysmorphologies [34]. In addition, the impaired proliferation and apoptosis of CNCCs could change the number of chondrocytes and further induce cartilage dysmorphologies. Cheng et al. [35] presented a model positing that a reduced number of CNCCs would contribute to their abnormal delamination and differentiation and that such abnormalities might induce chondrocyte misarrangement and cartilage dysmorphologies. Therefore, the defects in craniofacial cartilages in *vwa1* mutants appear to reflect impaired development of CNCCs.

CNCC development is governed by a hierarchically arranged gene-regulatory network controlled by several signaling pathways, including Wnt, bone morphogenetic protein (BMP), and FGF pathways [29]. In *vwa1* mutants, the expression of *fgf8a* and its receptors was decreased, indicating that FGF signaling might be downregulated. The FGF pathway contributes to chondrogenesis by regulating cell morphology, migration, survival, and proliferation [10] and by maintaining a balance in chondrocyte metabolism [36]. Thus, the downregulation of the FGF pathway in *vwa1* mutants might contribute to the defects in CNCC development and chondrogenesis in these mutants. The abnormal CNCC condensation observed in *vwa1* mutants was marked by decreased *barx1* expression. Similarly, in previous zebrafish models, mutation of *fgf8* or *fgfr1* and blockade of the FGF pathway was shown to cause a decrease in the expression of *barx1* and abnormal CNCC condensation [37,38]. A chick model has also demonstrated that FGF8 can substitute for epithelial signals to maintain the expression of *barx1*. [39]. In addition, the expression of *col2a1a* was also reduced in *vwa1* mutants, indicative of deficient CNCCs differentiation. This phenomenon was also reported in *fgf8a* mutant zebrafish [40], indicating that decreased expression of *fgf8a* might also contribute to the defects in CNCC differentiation in *vwa1* mutants. Other studies have reported that the FGF pathway stimulates proliferation and inhibits apoptosis by activating MAPK, PI3K-AKT, and STAT pathways [41,42]. Dexamethasone exposure in a chick model was shown to inhibit *Fgf8* expression in association with increased apoptosis of neural crest cells [35]. Therefore, the decreased proliferation and increased apoptosis observed in *vwa1*^−/−^ mutants might also reflect the downregulation of FGF signaling. Finally, the expression of *runx2a* also decreased in *vwa1*^−/−^ mutants, an effect that also seems to result from decreased FGF signaling, given that *runx2* is stimulated by FGF [41]. Therefore, dysregulation of FGF signaling may induce the abnormal development of CNCCs, and thereby cause the dysmorphologies of craniofacial cartilages.

Downregulation of the FGF pathway in *vwa1* mutants may be caused by several mechanisms. First, bioinformatic tools predicted that WARP interacts with several elements in the FGF pathway [12]; thus, *vwa1* genetic ablation may directly influence FGF signaling. Second, WARP expression is restricted to the ECM of chondrocytes, skeletal muscle, and peripheral nerves [19]. WARP protein also co-localizes and binds at a high affinity with collagen VI and perlecan in the articular cartilage matrix [43,44], thus bridging these molecules to stabilize the ECM structure [19]. Collagen VI plays an essential role in chondrogenesis by anchoring chondrocytes and mediating cell–matrix interactions [45]. Perlecan is a large heparan sulfate proteoglycan [46], and paracrine FGFs, including the FGF8 subfamily, can enhance ECM stability by binding heparan sulfates [47]. To date, perlecan has been reported to bind a number of FGFs, including FGF1, FGF2, FGF7, and FGF18, and promote the development of chondrocytes [46]. Consequently, disrupting the interaction between perlecan and collagen VI may result in abnormal ECM structure and decreased stability of FGFs, thereby hindering FGF signaling. Third, the decreased expression of *fgf8a* and its receptors in *vwa1* mutants (via as yet unclear mechanisms) would cause a decrease in the concentration of both growth factors and receptors, which would downregulate FGF signaling. Therefore, decreased expression of *vwa1* appears to affect FGF signaling through various mechanisms, resulting in abnormal CNCC development and impaired chondrogenesis.

However, in several FGF8 mutant zebrafish and mouse models, developmental defects of craniofacial cartilage are more distinct in derivatives of the first pharyngeal arch than those of the second arch [40,48]. In contrast, *vwa1*^−/−^ mutants also showed serious defects in ceratohyal and ceratobranchial cartilages, which are derivatives of the second to seventh pharyngeal arches. The difference might be contributed by the involvement of changes in signaling pathways other than FGF in *vwa1* mutants. Because the FGF pathway can crosstalk with other pathways, such as TGFβ and Wnt [10], knockout of *vwa1* may create a complicated dysregulation involving several pathways that cumulatively influence chondrogenesis. Therefore, further studies are required to better elucidate the pathways influenced by *vwa1*.

## 5. Conclusions

In conclusion, we reported that *vwa1*, the zebrafish ortholog of the HFM candidate gene, *VWA1*, is essential for zebrafish craniofacial cartilage development. *vwa1* knockout might cause downregulation of FGF signaling, which could contribute to the impaired condensation, differentiation, proliferation, and apoptosis of CNCCs in these mutants. The abnormal development of CNCCs appears to result in chondrocyte disarrangement and deformities of craniofacial cartilage. We used CRISPR/Cas9 to develop a *vwa1*^−/−^ zebrafish line to exclude the potential influence of morpholino and gRNA injection, further reinforcing the validity of our results.

## Figures and Tables

**Figure 1 genes-14-00838-f001:**
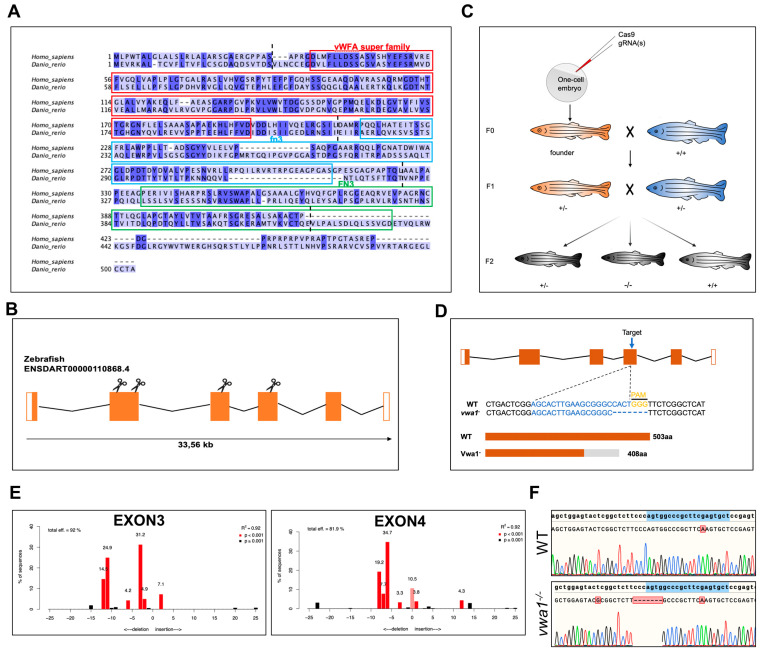
Conservation of *vwa1* across species and generation of *vwa1* knockout zebrafish by CRISPR/Cas9 genome editing. (**A**) Multiple sequence alignment of amino acids for human (445 aa) and zebrafish (503 aa) VWA1. Identical amino acids are shaded blue. The von Willebrand factor A-domain (vWFA), first fibronectin type III repeat domain (fn3), and second fibronectin type III repeat domain (FN3) are marked with red, blue, and green boxes, respectively. The boundaries between exons are marked with black dashed lines. (**B**) Structure of exons and introns of zebrafish *vwa1*. Untranslated regions are represented by white boxes, and coding regions are represented by orange boxes. Scissors indicate the position of the four CRISPR guides used to generate *vwa1*-KO crispants. (**C**) Stepwise construction of mutants. Adult F0 zebrafish were outcrossed with WT zebrafish, and their offspring were genotyped by PCR and Sanger sequencing to identify germline transmission of the *vwa1* deletion (F0 founders). A selected F0 founder was then crossed with WT zebrafish to obtain F1 heterozygotes, which were inbred to obtain F2 homozygous mutants. (**D**) Detailed illustration of mutant line generated by targeting exon 4. (**E**) Validation of the efficiency of gRNAs. Validation of the second gRNA targeting exon 2 failed owing to its intron-adjacent location. (**F**) Schematic diagram and sequencing results for WT controls and the *vwa1* mutant line in the F2 generation generated by CRISPR/Cas9 technology. Lower panel shows the target sequence (blue), protospacer adjacent motif (PAM) sequence (yellow), and 7-bp deletion in the mutant line.

**Figure 2 genes-14-00838-f002:**
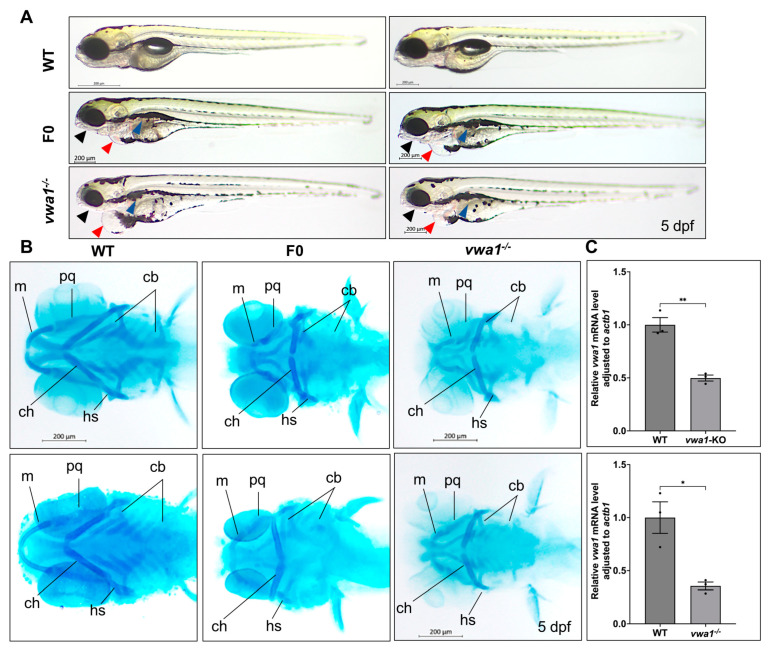
*vwa1* knockout in zebrafish causes craniofacial defects. (**A**) General phenotypic changes in F0 mosaic and F2 homozygous embryos. Morphology was assessed at 5 dpf; un-injected WT larvae were used as controls (WT). Black arrow, lower jaw underdevelopment. Blue arrow: altered swim bladder development. Red arrow: pericardial edema. (**B**) Cartilage formation at 5 dpf, detected by Alcian Blue staining. Compared to controls, F0 crispants and F2 homozygous larvae showed hypoplastic Meckel’s cartilage and palatoquadrate, a significantly widened angle between ceratohyals, and a reduction in the number of hypoplastic ceratobranchial cartilages. m, Meckel’s cartilage; pq, palatoquadrate; cb, ceratobranchial; ch, ceratohyal; hs, hyosymplectic. (**C**) RT-qPCR analysis of *vwa1* expression levels in F0 crispants and the −7 bp deletion mutant line at 5 dpf. n = 3 independent experiments. * *p* < 0.05; ** *p* < 0.01.

**Figure 3 genes-14-00838-f003:**
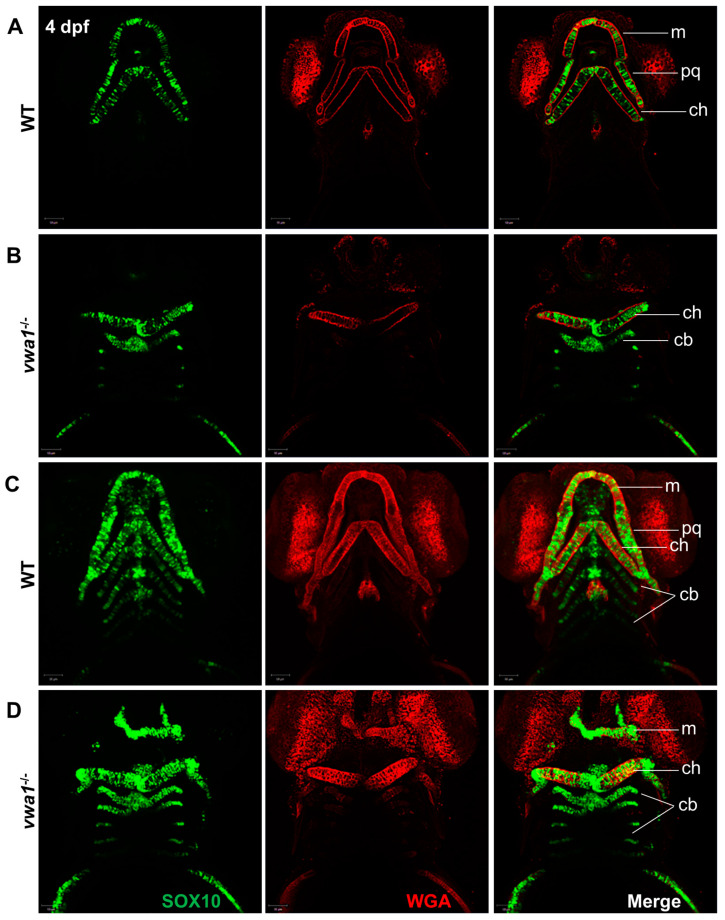
Immunofluorescence assay of WGA and SOX10 reveals a disorganized arrangement of chondrocytes in mutant embryos. (**A**,**B**) A WT control embryo at 4 dpf, with single-layer images (**A**) showing normal craniofacial cartilage morphogenesis and stacked images (**B**) showing the characteristic “stack of pennies” arrangement of cartilage cells in which the slender cartilage cells assembled with each other to form their own cartilage elements. (**C**,**D**) Compared with the WT control group, (**C**) single-layer and (**D**) stacked images showed deformities in the overall morphology of the mandibular cartilage of *vwa1* mutants, a near absence of Meckel’s cartilage and palatal square cartilage, severely deformed ceratohyal, and smaller size and aspect ratio of many chondrocytes. m, Meckel’s cartilage; pq, palatoquadrate; cb, ceratobranchial; ch, ceratohyal.

**Figure 4 genes-14-00838-f004:**
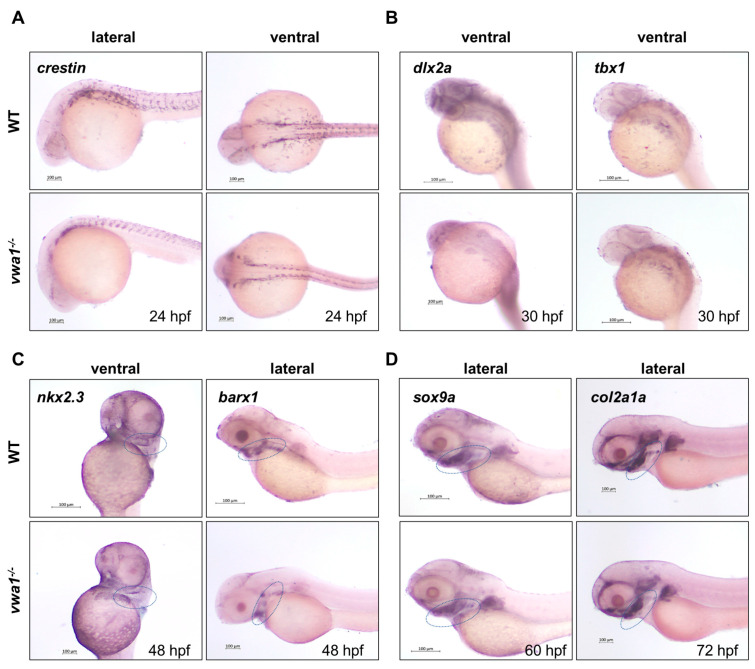
vwa1 functions in cranial neural crest cell condensation. (**A**) WISH analysis of *crestin* at 24 hpf showed no significant difference between *vwa1* mutants and WT controls. (**B**) The expression pattern of *dlx2a* and *tbx1* were essentially normal in the pharyngeal arch at 30 hpf, although the absolute area was smaller owing to the reduced body size of mutants. (**C**) At 48 hpf, expression of the endodermal pouch marker, *nkx2.3*, was not significantly different between mutant embryos and WT controls. However, the expression of *barx1*, indicative of the condensation of prechondral mesenchymal cells, was significantly reduced in the pharyngeal region of *vwa1* mutants compared with that in WT controls. (**D**) Expression level of *sox9a* did not change significantly at 60 hpf, whereas the expression of *col2a1a* decreased at 72 hpf. The mandibular region is marked with dotted blue circles.

**Figure 5 genes-14-00838-f005:**
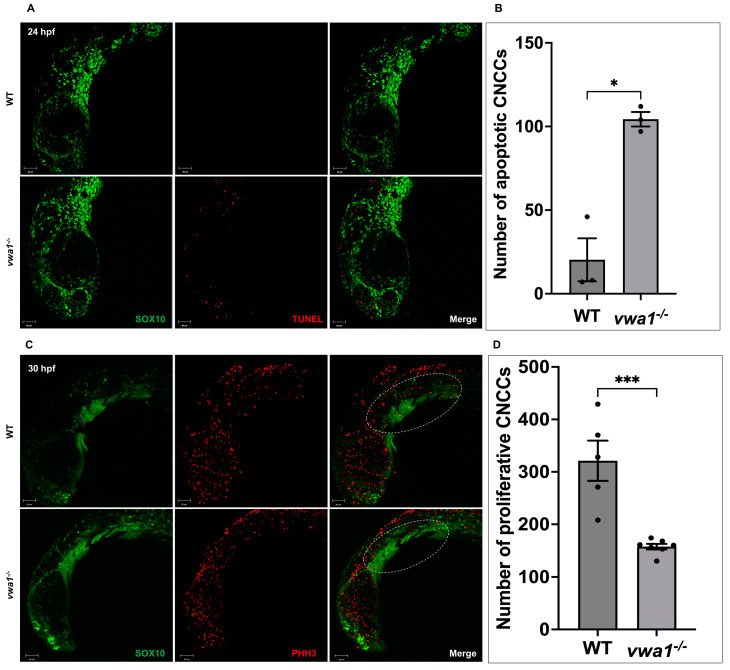
Increased apoptosis and decreased proliferation of cranial neural crest cell in *vwa1* mutant embryos. (**A**,**B**) TUNEL staining (red fluorescence) revealed increased cell apoptosis in the cranial region and dorsal tissues of *vwa1* mutant embryos at 24 hpf. SOX10 marked CNCCs. (**C**,**D**) PHH3 staining (red fluorescence) showed decreased cell proliferation in the cranial region of *vwa1* mutant embryos at 30 hpf. SOX10 marked CNCCs. The pharyngeal arches are marked with dotted white circles. * *p* < 0.05; *** *p* < 0.001.

**Figure 6 genes-14-00838-f006:**
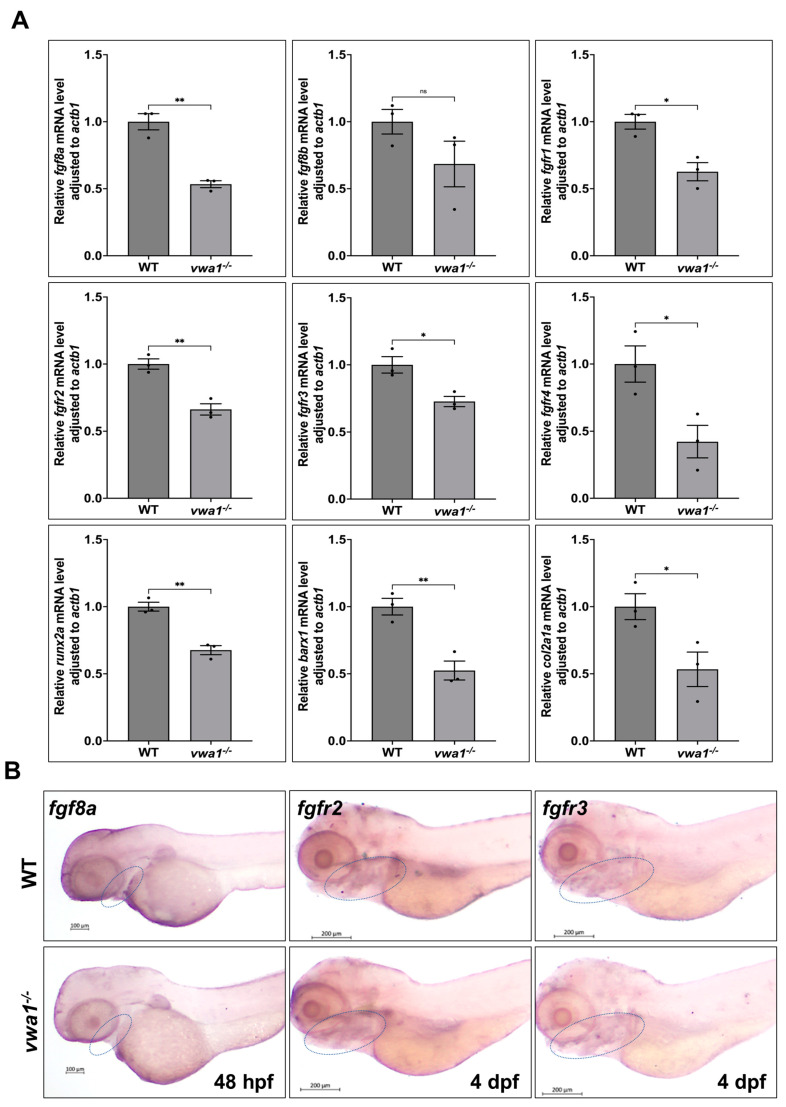
vwa1 regulates FGF signaling in the pharyngeal region. (**A**) RT-qPCR analysis of the mRNA levels of *fgf8a*, *fgf8b*, *fgfr1*, *fgfr2*, *fgfr3*, *fgfr4*, *runx2a*, *barx1*, and *col2a1a* in *vwa1*^−/−^ zebrafish, showing that all of these FGF signaling pathway components except for *fgf8b* were decreased in *vwa1* mutants. Relative transcription levels were calculated as fold change using the 2^−ΔΔCt^ method. (**B**) WISH staining of *fgf8a*, *fgfr2*, and *fgfr3* in *vwa1* mutant embryos and WT controls showed that *fgf8a*, *fgfr2*, and *fgfr3* expression were all reduced in mutant embryos. The mandibular region is marked with dotted blue circles. Appendix A Apoptosis did not significantly increase in *vwa1*^−/−^ mutants at 30 and 48 hpf. (**A**,**B**) TUNEL assay (red fluorescence) at 30 hpf demonstrated similar amount of apoptosis cells in the cranial region and dorsal tissues of wildtype controls (WT) and mutants. (**C**,**D**) TUNEL assay (red fluorescence) at 48 hpf showed similar level of apoptosis in WT and mutants. * *p* < 0.05; ** *p* < 0.01.

## Data Availability

All figures are associated with raw data. Raw images can be provided upon request. Sequences used for CRISPR/Cas9 and for in situ hybridization are included in the Appendix A.

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
