# Peer review of "vwa1 Knockout in Zebrafish Causes Abnormal Craniofacial Chondrogenesis by Regulating FGF Pathway"

_genes, 2023, doi:10.3390/genes14040838_

Round 1

Reviewer 1 Report

The cause of hemifacial microsomia (HFM), the second most common congenital craniofacial defect after cleft lip and palate, is still uncertain. In the manuscript, Niu et al. have described the role of VWA1, a novel candidate gene for HFM, in craniofacial cartilage development. I do not have any major comments on the research design, results presentation and conclusions – the manuscript is well written.

However, I have found some similarities between the current and previous paper published by the research team in Front. Cell Dev. Biol. (09 September 2020, Sec. Epigenomics and Epigenetics, Volume 8 – 2020; https://doi.org/10.3389/fcell.2020.571004).

 In the revised manuscript, there is information that “morpholino used by our previous study [9] to knockdown vwa1 in zebrafish was inevitably limited by the off-target effect, short maintenance time, and non-specific activation of P53 pathway. ……. Therefore, in the current study, we generated vwa1 knockout zebrafish by CRISPR/Cas9 technique to further study the function of vwa1 in craniofacial cartilage development”. However, in the cited paper, the vwa1 gene in zebrafish was also knocked out using the CRISPR/Cas9 system.

 In addition, Whole-Mount In situ hybridization results are presented in both papers (genes analyzed in the current paper: crestin, dlx2a, tbx1, barx1, nkx2.3, fgf8a, sox9a, and col2a1a; genes analyzed in previous paper: crestin, dlx2a and sox9a). Some results presented in the figures also seem to be similar.

Can the Authors comment on my remarks? The manuscript should present only new results; if there are some repetitions, they should be omitted, or there should be a clear statement that the results have already been published.

 It would also be valuable to discuss the power and strengths of zebrafish models for understanding craniofacial anomalies.

Reviewer 2 Report

Manuscript ID: genes-2276560

Title: vwa1 knockout in zebrafish causes abnormal craniofacial chondrogenesis by regulating FGF pathway

Authors: Xiaomin Niu, Fuyu Zhang, Lu Ping, Yibei Wang, Bo Zhang, Jian Wang, and Xiaowei Chen

In this study, Niu and colleagues studied the function of the vwa1 gene, which has been linked to Hemifacial microsomia in previous studies. The authors took a novel approach by using CRISPR/Cas9 to knock out the gene in the zebrafish model. The resulting mutants showed cartilage dysmorphologies including hypoplastic Meckel’s cartilage and palatoquadrate cartilage,  as well as malformed ceratohyal and ceratobranchial cartilages.

Through a variety of molecular methods, the authors found that the resulting mutants  exhibited abnormal cranial neural crest cell (CNCC) condensation and differentiation, as well as cell death. The manuscript is clearly written and appears to be scientifically sounds. However, some revision, particularly having to do with figures, is needed before the manuscript is ready for publication. The list below is in no particular order or importance:

1. It would be helpful to include the exon boundaries in Figure 1 Panel-A, if not the CRISPR/Cas9 sites even.

2. In Figure 2B, the vwa1-/- images appear overexposed. There does appear to be difference in cartilage morphology, as the authors suggest, however it is very difficult to see in the example images of the figure. The Figures should be replaced with some which are comparable to the WT and F0 samples.

3. In Figure 3, the authors should consider labeling the individual cartilages (e.g. Meckel’s, palatal square, etc.) so that the reader can easily see what the authors are referring to.

4. In lines 265 - 267, the authors write “Then, CNCCs migrate to the pharyngeal arches, differentiate into pharyngeal arch primordia at 30 hpf, and eventually become mature. The primordia  can be marked by dlx2a and tbx1, while the mature CNCCs can be marked by nkx2.3.”
The authors should modify the text to write exactly what cell or tissue in the primordia dlx2a and tbx1 mark, especially since they do this for nkx2.3. 

5. In lines 273-275, the authors write “The expression 273 area of dlx2a and tbx1 in mutants was smaller in absolute terms, but the expression pattern was conserved relative to their smaller body size (Figure 4B).”.
This is the first instance of the authors discussing a smaller body size of the mutants. This is an important feature that should be mentioned earlier in the manuscript as a part of the general phenotype of the mutants.

6. In lines 279-21, the authors write “Furthermore, in the vwa1 mutants, the expression of sox9a was similar with controls, while the expression of col2a1a was significantly reduced.”
In looking at the images in Figure 4D, the expression of col2a1a does not appear to be significantly reduced at all. In fact, one could argue that Sox9 shows a reduced expression in comparison. If the authors are going to make the claim of reduced col2a1a, they need to either have different, images where the differences is clear, or else maybe use arrows to pinpoint the differences, because at a general glance, it is not apparent at all.

7. Figure 5A - 24hr  vwa1-/- TUNEL image does not appear to show any significant signal. The image needs to be replaced with one that is brighter. I can see very faint signal if I zoom in very close, but otherwise this particular image is not useful as a manuscript figure since it is so dark. 

8. Figure 5A, there appears to be a difference in the Sox10 expression between the WT and  vwa1-/-. Is this real? if not, why does it appear darker in the mutant?

9. Figure 5A and C - The authors should label the parts of the fish once again. It appears that an eye is visible, is this correct? it is generally difficult to tell where the expression of the markers are.

10. In lines 301-302, the authors write “In contrast, TUNEL showed absence of apoptotic cells in the pharyngeal region of vwa1 crispants and mutants at 30 hpf and 48 hpf.”. However, there is no image of TUNEL analysis in Figure 5C. Was this done but not included? If so, the authors should write “Data not shown” or some  other version of this. However, if they have done an experiment, they should strongly consider including it.

11. Regarding Figure 5C-lines 304-306, the authors write “PHH3 staining revealed notably reduced mitotic PHH3-positive CNCCs in vwa1 mutant embryos than those in control embryos at 30 hpf,indicating that the proliferation of CNCCs were inhibited (Figure 5C, D).”
However, the animals in the WT and vwa1-/- appear to be different sizes, so how did the authors do the cell counts? Is this something that needs to be considered? Should the cell counts be relative to size or image area?

12. In lines 324-327, the authors write “We found the expression of fgf8a, FGF receptors including fgfr1, fgfr2, fgfr3, fgfr4, and downstream signaling molecule runx2a were decreased in the vwa1-/- zebrafish embryos, and the expression of fgf8b was not significantly different (Figure 6A)." However, fgf8b analysis does not appear to be part of Figure 6 A at all. Once again the authors appear to have done an experiment but left it out. If it is going to be mentioned in the manuscript, the result should be included, or else clearly state that data was not included in the manuscript. Otherwise it is confusing for the reader as we go searching for a result that is not there. 

 13.  In lines 324-327, the authors also state that they found the expression of fgf8a was significantly decreased in the vwa1-/- zebrafish embryos. Yet, in Figure 6A, the graph for fgf8a does not show an asterisk, which to me would indicate no significance.

Does the lack of an asterisk indeed mean that the difference is not significant? The authors should first, clearly state what the asterisks mean, and also clarify why fgf8a does not show an asterisks but is still considered significantly decreased.

14. Regarding the graphs in Figure 6C, to the best of my knowledge, whole-mount RNA in situ hybridization is not quantitative or is semi-quantitative at best. However, if the authors are going to quantitate the results, the detailed methods for how quantitation was done needs to be included in the methods section.

15.  In lines 324-327, the authors write “Therefore, the defects of vwa1 mutants may be predominantly attributed to the decrease in vwa1 expression.” The authors should clarify why they believe there was a decrease in expression. What was the cause?

16. In lines 437, the authors write “However, targeted rescue against FGF pathway was unsuccessful…” did the authors attempt this as well? It does not seem to be part of the manuscript.

Once again, if experiments have been done, they either need to be stated clearly that they have been done and not included or else left out entirely. The authors should not discuss experiments in the discussion which were not presented.

Round 2

Reviewer 2 Report

The authors have addressed all of my concerns.

I just have one minor comment. In addressing my concern about the size difference in animals, the authors use the term "numerically smaller", which may sound like smaller in number and not size.

I would suggest that the authors either just say "smaller" or "smaller in size", as saying "numerically smaller" risks confusing the reader.
